# High Arterial Lactate Levels after Hepatic Resection Are Associated with Low Oxygen Delivery and Predict Severe Postoperative Complications

**DOI:** 10.3390/biomedicines10051108

**Published:** 2022-05-10

**Authors:** Rita Gaspari, Luciana Teofili, Francesco Ardito, Enrica Adducci, Maria Vellone, Caterina Mele, Nicoletta Orlando, Tiziana Iacobucci, Massimo Antonelli, Felice Giuliante

**Affiliations:** 1Dipartimento di Scienze dell’Emergenza, Anestesiologiche e della Rianimazione, Fondazione Policlinico Universitario A. Gemelli IRCCS, 00168 Rome, Italy; rita.gaspari@unicatt.it (R.G.); enrica.adducci@unicatt.it (E.A.); tiziana.iacobucci@policlinicogemelli.it (T.I.); massimo.antonelli@unicatt.it (M.A.); 2Dipartimento di Scienze Biotecnologiche di Base, Cliniche Intensivologiche e Perioperatorie, Università Cattolica del Sacro Cuore, 00168 Rome, Italy; 3Dipartimento di Diagnostica per Immagini, Radioterapia Oncologica ed Ematologica, Servizio di Emotrasfusione, Fondazione Policlinico Universitario A. Gemelli IRCCS, 00168 Rome, Italy; luciana.teofili@unicatt.it (L.T.); nicolettaorlando@hotmail.it (N.O.); 4Dipartimento di Scienze Radiologiche ed Ematologiche, Università Cattolica del Sacro Cuore, 00168 Rome, Italy; 5Dipartimento di Scienze Mediche e Chirurgiche, Chirurgia Generale ed Epato-Biliare, Fondazione Policlinico Universitario A. Gemelli IRCCS, 00168 Rome, Italy; maria.vellone@unicatt.it (M.V.); caterina.mele2@gmail.com (C.M.); felice.giuliante@unicatt.it (F.G.); 6Dipartimento di Medicina e Chirurgia Traslazionale, Università Cattolica del Sacro Cuore, 00168 Rome, Italy

**Keywords:** hepatectomy, hyperlactatemia, anemia, oxygen delivery, hemodynamic monitoring, postoperative complications

## Abstract

High End-Surgery Arterial Lactate Concentration (ES-ALC) predicts poor outcome after hepatectomy. The aim of this study was to identify intraoperative hemodynamic parameters predicting high ES-ALC during elective liver resection. Patients who underwent liver resection between 2017 and 2018, under FloTrac/EV1000^TM^ hemodynamic monitoring, were included. The ES-ALC cutoff best predicting severe postoperative complications was identified. Association between high ES-ALC and preoperative and intraoperative variables was assessed. 108 patients were included; 90-day mortality was 0.9% and severe morbidity 14.8%. ES-ALC cutoff best discriminating severe complications was 5.05 mmol/L. Patients with ES-ALC > 5.0 mmol/L had a relative risk of severe complications of 2.8% (*p* = 0.004). High ES-ALC patients had longer surgery and ischemia duration, larger blood losses and higher requirements of fluids and blood transfusions. During surgery, hemoglobin concentration and oxygen delivery (DO_2_) decreased more significantly in patients with high ES-ALC, although they had similar values of stroke volume and cardiac output to those of other patients. At multivariate analysis, surgery duration and lowest recorded DO_2_ value were the strongest predictors of high ES-ALC. ES-ALC > 5.0 mmol/L in elective liver resection predicts postoperative morbidity and is essentially driven by the impaired DO_2._ Timely correction of blood losses might prevent the ES-ALC increase.

## 1. Introduction

An increase of arterial lactate concentration (ALC) after surgery is a poor prognosis predictor in various clinical settings [1]. The term hyperlactatemia usually refers to an ALC between 2 and 5 mmol/L, with severe hyperlactatemia usually indicating ALC beyond 5 mmol/L [1]. The increase of ALC intraoperatively or at the end of surgery is associated with higher morbidity and mortality rates also in elective hepatic resection [2,3,4,5,6,7,8]. At variance with other settings, however, the exact ALC threshold predicting a poorer outcome in patients with liver resection still remains to be defined.

Among various organs and tissues, the liver has the greatest ability to metabolize lactate, while the kidneys, skeletal muscle, and heart contribute to lactate clearance [1]. Different mechanisms underlie the compromised lactate metabolism occurring in liver resection. First, intraoperative surgical and anesthetic techniques usually adopted to minimize blood losses can themselves induce an ALC rise [9,10,11]. Actually, the intermittent clamping of the hepatic pedicle during parenchymal resection induces ischemia and ischemia-reperfusion of hepatic cells, thus decreasing the liver ability to metabolize blood lactate [10,11] In addition, the liver ischemia per se results in a significant cytokine release, finally leading to an increased lactate production [12]. This increase is also influenced by the extension of liver resection and/or underlying hepatic diseases [13]. Likewise, current anesthetic techniques for reducing intraoperative bleeding usually aim to maintain a low central venous pressure (CVP) through a restrictive regimen of fluid therapy that further promotes local and systemic hypoperfusion [14,15]. Similarly, hypoperfusion may be induced by the administration of diuretics or vasodilators, epidural anesthesia, or volatile anesthetics with a vasodilator effect [9].

The non-invasive hemodynamic monitoring techniques have entered clinical practice to optimize the management of intraoperative fluid therapy. One of the most largely used is the FloTrac/EV1000^TM^ (Edwards Life Sciences, Irvine, CA, USA, henceforth FloTrac), which measures the stroke volume variation (SVV) as an indicator of the intravascular volume filling, and fluid reactivity [16,17,18,19,20,21]. To date, the SVV measurement has proved to be a valid alternative to CVP measurement also in in liver surgery [20,21]. However, whether non-invasive hemodynamic monitoring may provide information on the source of hyperlactatemia during hepatectomy remains to be investigated.

The aim of this study was to assess whether hemodynamic parameters recorded at FloTrac monitoring predict high end-surgery ALC (ES-ALC) in elective liver resection.

## 2. Materials and Methods

### 2.1. Study Design

This was a retrospective analysis of prospectively maintained database of adult patients (aged ≥ 18 years) undergoing hepatic resection at the Hepatobiliary Surgery Unit, at Fondazione Policlinico Universitario A. Gemelli IRCCS of Rome, between January 2017 and December 2018. The study inclusion criterion was the availability of parameters recorded at hemodynamic monitoring through the FloTrac/EV1000^TM^ (version 1.9, IV generation; Edwards Lifesciences Corp. One Edwards Way, Irvine, CA 92614, USA). Exclusion criteria were simultaneous colorectal and hepatic resection and concomitant renal failure (serum creatinine level > 1.3 mg/dL). During the study period, liver surgery was performed by the same team of hepatobiliary surgeons and anesthesiologists and managed according to standardized procedures and decisional algorithms.

### 2.2. Anesthetic Management

Anesthesia was induced using intravenous propofol (2 mg/kg), fentanyl (3 µg/kg), and rocuronium (0.6 mg/kg) for muscle relaxation. After endotracheal intubation, the anesthesia was maintained with sevoflurane 2% with MAC 1, and intravenous remifentanil infusion (dose ranging from 0.08–0.1 µg/kg/min). All patients were ventilated with a mixture of air and oxygen, with a tidal volume of 8 mL per kg of the ideal body weight, to maintain the end-tidal carbon dioxide at 35 to 45 mmHg, and peripheral oxygen saturation (SpO_2_) > 97%. Continuous standard monitoring included electrocardiogram (D2-V5), heart rate (HR), SpO_2_, and invasive arterial blood pressure. At different times before and during surgery, arterial blood samples for gas analysis were collected in heparinized syringes at 37 °C and tested using a Nova Stat Profile blood gas analyzer (Nova Biomedical Corporation, Waltham, MA, USA). Hemodynamic monitoring was carried out using the FloTrac/EV1000^TM^ system, as detailed previously [16,17,18,19,20,21]. Briefly, an arterial catheter was placed in the radial artery, and SVV was estimated in mechanically ventilated patients by arterial pulse contour analysis. The SVV was expressed as the percentage of the starting stroke volume (SV), with values greater than 13–15% predicting the ability of the left ventricle to increase the SV and cardiac output (CO) in response to fluid administration [16,17,18,19]. Oxygen delivery (DO_2_) was calculated with the FloTrac system by entering hemoglobin (Hb) concentration and arterial oxygen saturation (SaO_2_) recorded at blood gas analysis. The formula for DO_2_ calculation was DO_2_ = CO × (1.39 × Hb × SaO_2_) + (PaO_2_ × 0.003), where PaO_2_ indicates the arterial oxygen partial pressure. The target SVV in our patients was 13–15% during the hepatic resection phase and ≤10% at completed hepatic dissection. Intraoperative fluid therapy was restricted at less than 5 mL/kg/h for the entire duration of surgery, although it was minimized during the hepatic resection phase and accelerated once the resection was completed. If the surgery was performed through laparoscopy, the intra-abdominal intraoperative pressure was maintained at 12 mmHg in order not to affect the hemodynamic measurements of the FloTrac.

Preconditioning, together with systematic hemodynamic monitoring with the Flotrac device, and intraoperative monitoring of urinary output and of temperature were always performed in order to guarantee patient homeostasis. Electrolyte changes were systematically corrected in all patients. Crystalloid (balanced saline solution at pH 7.4) and colloid (5% albumin) were used. Noradrenaline was administered if the mean arterial pressure (MAP) dropped below 65 mmHg. When the urine output was <20 mL/h, additional 200 mL of fluids in 10 min were quickly infused. Packed red blood cells (RBCs) were transfused at Hb concentration below 8 g/dL. Correction of metabolic acidosis due to hyperlactatemia was treated with sodium bicarbonate in case of pH values < 7.2.

In the postoperative period, pH return to normal values (pH > 7.35) was obtained through the optimization of fluid therapy with balanced saline solutions in order to allow a gradual recovery of the hypoperfusion that was induced during surgery.

At end of surgery, patients were extubated and transferred to general ward, except for those with a history of ischemic heart disease or those receiving extensive and long-lasting (≥360 min) hepatic resection, who were admitted to the intensive care unit (ICU).

This type of anesthetic management was part of our Enhanced Recovery After Surgery (ERAS) program, introduced in our Unit in 2016. This program was approved, on 15 September 2016, by the Ethics Committee of the Catholic University of the Sacred Heart of Rome (Prot. n. 33896/16; ID: 1326).

### 2.3. Surgical Procedure

Liver resection was defined according to the International Hepato-Pancreato-Biliary Association (IHPBA) terminology [22]. Resections of ≥3 segments were classified as major hepatectomies. Multiple resections included patients undergoing ≥3 parenchymal sparing liver resections for liver metastases and were classified as minor complex hepatectomies. The surgical technique used in our unit for liver resection has been described previously [23,24,25]. Briefly, parenchymal transection was performed with the Cavitron ultrasonic surgical aspirator (CUSA 200; Valleylab, Boulder, CO, USA) and wet bipolar forceps; hemostasis and biliostasis were obtained with absorbable clips (Absolok AP200 and AP300, Ethicon, Johnson & Johnson Medical Devices Companies, Pratica di Mare, Pomezia, Rome, Italy) or with 3/0–4/0 absorbable stitches and unabsorbable ones on hepatic veins branches. Intermittent hepatic pedicle clamping was not routinely started at the beginning of liver resection but was used only when bleeding was hindering a clear view of the operative field. The program for minimally invasive liver surgery started in our Unit in 2009 [26]. In the case of minimally invasive liver surgery, the patient was placed in supine position or middle left lateral position according to the tumor location, with the surgeon between the legs. Five trocars were usually inserted [26]. Liver resection was carried out by the 80-degree articulating vessel sealer (Aesculap Caiman; B. Braun, Tuttlingen, Germany) and by CUSA. Some of the investigated patients were included in Enhanced Recovery After Surgery (ERAS) program, first described by Kehlet et al. for colorectal surgery in 1997 [27] and introduced in our Unit in 2016 [28].

### 2.4. Collected Variables and Definitions

Epidemiologic and clinical data included: age, sex, body mass index (BMI), inclusion in the ERAS program, American Society of Anesthesiology physical (ASA) [29] class, comorbidities (diabetes, arterial hypertension, coronary artery disease, and mild chronic obstructive pulmonary disease), laparoscopy or laparotomy approach, extent of liver resection, cirrhosis or steatosis above 25% (documented at histological examination of resected liver), operation time, overall Pringle’s maneuver duration (if any), total intravenous fluid infusion, urine output, blood losses, noradrenaline administration, and RBC unit transfused during surgery and until discharge. Biochemical and hemodynamic variables included ALC, blood glucose level, Hb concentration, SaO_2_, actual base excess (BE), arterial pH, SV, SVV, CO, mean arterial pressure (MAP), and DO_2_. All data were recorded before starting surgery (after anesthesia induction and radial artery cannulation), at the completion of surgery (immediately prior to the abdominal wall closure), and before and after every Pringle’s maneuver, if any.

Morbidity was defined as the occurrence of any clinical complication within 30 postoperative days. Complications were classified according to Clavien–Dindo classification [30] and patients were grouped as having mild (I and II) or severe (IIIa to V) complications. Mortality was defined as death for any cause occurring within 90 days from liver resection. High end-surgery (ES) ALC was defined according to the ES-ALC value best discriminating severe day-30 morbidity.

### 2.5. Study Outcomes

Primary endpoint: to identify intraoperative hemodynamic parameters predicting high ES-ALC.

Secondary endpoint: to evaluate the correlations between high ES-ALC and postoperative severe complications.

### 2.6. Statistical Analysis

Continuous variables were expressed as median (interquartile range, IQR) and categorical variables as *n*, (%). The ES-ALC cut off best discriminating the occurrence of severe complications was identified by Receiving Operating Characteristics (ROC) curve analysis according to the highest sensitivity and specificity. The ES-ALC value was then used to categorize patients into high- or low-ES-ALC groups at subsequent univariate and multivariate analysis. Differences between groups at univariate analysis were assessed by the Mann–Whitney U test for continuous variables and the Fisher’s exact test for categorical variables. The Wilcoxon test for paired samples was used to assess differences among values of the same variables recorded at different times. Linear regression analysis was used to predict ES-ALC according to other parameters. The combined effect of different variables on ES-ALC was evaluated in multivariate logistic regression analysis models using the backward stepwise method. Those variables significantly different at univariate analysis (*p* < 0.05) and any further variable with a reasonable effect on the outcome were included among covariates. The test of Hosmer and Lemeshow was used to assess the quality of models. Analyses were performed using the IBM SPSS Statistics for Windows (Version 27.0. IBM Corp. Released 2020, Armonk, NY, USA) and GraphPad Prism (version 6.00 for Windows, GraphPad Software, La Jolla, CA, USA). The data supporting the findings of this study are available from the senior author upon reasonable request.

## 3. Results

Between January 2017 and December 2018, 340 patients underwent liver resection at our Unit. Out of the 340 patients, data from hemodynamic monitoring with FloTrac were available for the analysis in 111. Three patients were excluded because of simultaneous colorectal resection, and data recorded from 108 patients were included in the analysis. Clinical characteristics and surgery features of the study population are reported in Table 1. Fluid balance and hemodynamic parameters are reported in Table 2. In total, 16 patients (14.8%) developed severe complications, and 1 patient died (0.9%).

### 3.1. Lactate Concentration and Risk of Complications

In ROC analysis, the ES-ALC cutoff value of 5.05 mmol/L best predicted the risk of severe complications, with an AUC of 0.661 ± 0.079 (95% CI 0.506–0.816), a sensitivity of 56.2%, and a specificity of 80.4% (Figure 1). Twenty-seven patients (25%) had ES-ALC > 5.0 mmol/L, with values ranging from 5.1 to 10.6 mmol/L. The complication rate was 33.3% (95% CI 16.5–53.9) among high-ES-ALC patients and 8.6% (95% CI 3.5–16.9) in patients without high ES-ALC, with a relative risk of 2.8% (95% CI 1.6–5.2) for the former ones (*p* = 0.004). The rate of ICU admission was similar between groups (40.7% versus 25.9%, *p* = 0.131).

### 3.2. Clinical Characteristics and Laboratory Parameters Associated with High End-Surgery Arterial Lactate Concentration

Differences in clinical characteristics and hemodynamic parameters of patients grouped according to the ES-ALC cut-off are illustrated in Table 3 and Table 4, respectively. Groups were comparable at baseline for clinical characteristics and blood test variables, except for BMI, prevalence of hypertension (Table 3), and pre-operative ALC (Table 4) that were higher in patients with high ES-ALC. During surgery, patients with high ES-ALC more frequently underwent extensive liver resections, with longer duration of surgical procedure and Pringle’s maneuver (Table 3). Likewise, high ES-ALC patients showed larger blood losses, were given more fluids and RBCs transfusions, had lower urine output, and more frequently required noradrenaline administration. The Hb concentrations recorded either intraoperatively (lowest value) or at the end of surgery were lower in ES-ALC patients, in association with higher HR, lower BE, and arterial pH values. In this series, although the group of patients with ES-ALC values > 5.0 mmol/L had significantly lower pH and more negative BE than patients with ES-ALC ≤ 5 mmol/L, none of the patients presented pH < 7.2 (Table 4).

The correlation between ES-ALC and postoperative complications is illustrated in Table 5: the rate of severe complications (Clavien–Dindo grade ≥ 3) was significantly higher in high ES-ALC patients than in low ES-ALC patients.

At linear regression analysis, the ES-LAC levels significantly depended from the magnitude of Hb reduction, blood loss, volume of replaced fluids, and surgery duration (Figure 2). ES-ALC groups were substantially comparable for the assessments of MAP, SVV, SV, and CO, at the start, during (lowest value), and at the end of surgery. In contrast, the assessments of DO_2_, either intraoperative (lowest value) and at the end of surgery, appeared significantly lower in the high ES-ALC group (Table 4).

On the basis of these results, different models of multivariate logistic regression analysis were designed to predict high ES-ALC. Age, BMI, surgery duration, Pringle’s maneuver duration, blood loss, Hb concentration (either lowest or end-surgery), and DO_2_ (either lowest or end-surgery) were included as continuous variables, while diabetes, arterial hypertension, coronary heart disease, cirrhosis, and major/minor complex resection were incorporated as categorical covariates. In multivariable logistic regression analysis, surgery duration and DO_2_ levels significantly predicted high ES-ALC, with an OR = 1.011 (95% CI 1.005–1.018, *p* = 0.001) for surgery duration, and OR = 0.989 (95% CI 0.979–0.999, *p* = 0.028) for lowest DO_2_ (costant = 0.013; Hosmer–Lemeshow *p* = 0.985) (Table 6).

### 3.3. Determinants for the DO_2_ Decrease in Patients with High End-Surgery Arterial Lactate Concentration

To further deepen mechanisms underlying the DO_2_ reduction in high ES-ALC patients, we compared DO_2_ modifications occurring during surgery in patients with and without high ES-ALC. We found that in patients with high ES-ALC, at variance with other patients, the lowest DO_2_ values did not recover to the initial levels (Figure 3). In a steadily ventilated patient, the level of DO_2_ basically depends on the CO value (i.e., SV per HR), and Hb concentration. Despite a persistent reduction of SV, in patients with high ES-ALC, a pronounced HR raise conceivably mediated the recovery of CO. Nevertheless, this was not paralleled by the recovery of the Hb concentration (Figure 3). Indeed, unbalanced blood losses occurring in high ES-ALC patients seemed to be the main drivers for the persistent low DO_2_ recorded at the end of surgery.

## 4. Discussion

The first finding of this study is that the arterial lactate concentration higher than 5.0 mmol/L at the completion of liver resection is associated with an increased risk of severe complications, finally resulting in a longer hospital stay. Notably, through the analysis of non-invasive hemodynamic records, we could definitely connect the hyperlactatemia of our patients to an impaired oxygen delivery resulting from the unbalanced blood loss. 

Several authors have shown that high ALC at the end of hepatectomy or in the early postoperative phase can be variably associated with renal and hepatic dysfunctions, postoperative peaks in serum bilirubin concentration, length of hospital stay, and mortality [2,3,4,5,6,7,8]. At variance with previous studies, we reported similar findings in a series of patient with a very low postoperative mortality rate. Watanabe et al. in 2007 examined for the first time the relationships between lactate level and outcome in liver resection [2]. The authors observed that arterial lactate levels were higher in non-survivors than in survivors (4.1 mmol/L versus 10.1 mmol/L) and in patients with complications (5.5 mmol/L vs. 3.6 mmol/L in other patients). Similarly, Wiggans et al. reported a significantly higher rate of renal failure and mortality in patients with a postoperative ALC ≥ 6.0 mmol/L in comparison with those with lactate levels ≤ 2 mmol/L [3]. More recent studies in the same surgical setting have attempted to identify an exact threshold of ALC predicting a poorer outcome. Vibert et al. reported that values ≥ 3.0 mmol/L predicted severe morbidity and mortality [4]. The inclusion of this criterion in a prognostic model greatly increased its accuracy, sensitivity, and specificity [4]. Similarly, Lemke et al. suggested that a lactate value ≥ 3.8 mmol/L was associated with an adverse outcome [6]. Finally, Niederwieser et al., evaluating lactate dynamics after hepatectomy, showed that the maximum level in the first 24 h is a robust indicator of poor outcome [8]. The authors demonstrated in an explorative cohort that an ALC ≥ 50 mg/dL (i.e., ≥5.5 mmol/L) was associated with a liver failure rate of 50%, an incidence significantly higher than that observed in patients with a maximum level below 50 mg/dL [8]. The authors confirmed this observation in a validation cohort and accordingly introduced this parameter in a prognostic score for the post hepatectomy liver failure [8]. At variance with the study of Niederwieser et al., our study did not examine the ALC dynamics during surgery but only value at baseline and the completion of surgery. Nevertheless, the ES-ALC cutoff predicting severe complications in our patients closely parallels that reported by Niederwieser et al. [8].

The raise of lactate levels after liver resection can be prompted by different underlying conditions and through different mechanisms. For example, comorbidities such as diabetes mellitus or concomitant cirrhosis may restrain the lactate clearance [3,4,5,8]. In our study, patients developing end-surgery hyperlactatemia had higher baseline ALC, higher BMI, and more frequently suffered from arterial hypertension. It could be hypothesized that these findings underlie a patients’ fragility, which overtly manifests only after the surgical stress. In agreement with our observation, other authors have reported that pre-operative ALC values ≥1.2 mmol/L predicts morbidity in patients submitted to liver resection, whereas values ≥4.5 mmol/L predicts mortality in the same population [13]. 

As previously reported, also in the present study, surgery duration was significantly associated with the raise of end-surgery lactate levels. Indeed, at the multivariable logistic regression analysis, duration of surgery and lowest DO_2_ were the strongest predictors of high ES-ALC. Conceivably, the time required for the surgical procedure otherwise reflects the hepatectomy complexity, and it also correlates with the overall duration of the Pringle’s maneuver. In fact, the alternating phases of ischemia and reperfusion, stimulate the release of cytokines activating Kuppfer cells to produce lactate [12]. Although other authors reported a link between hyperlactatemia and blood loss, to our knowledge, no studies have explored the role of hemodynamic variables in the hyperlactatemia development [3,4,6]. Using the non-invasive hemodynamic monitoring FloTrac system, we could highlight the relevant role played by the inadequate oxygen supply in hyperlactatemia development. Overall, we clearly demonstrated that although the cardiac output remained stable in all patients during surgery, ES-ALC increased in consequence of the Hb drop, which was not corrected by RBC transfusions. Conceivably, in patients with high ES-ALC, a critical tissue hypoxia might have occurred, progressively shifting metabolism towards anaerobiosis, with lactate generation. The reduced arterial pH levels seem to confirm this hypothesis. Nevertheless, the unavailability of data relative to the effective oxygen consumption, usually measured through the central venous oxygen saturation monitoring, makes this hypothesis only speculative.

Overall, our observations raise the issue of which hemoglobin thresholds should be set for transfusing patients undergoing liver resection. On one hand, RBC transfusions are alleged for increasing cancer recurrence [24,31]. On the other, it should be noted that 7 out 27 patients developing hyperlactatemia received transfusions during the early postoperative stay: among them, 5 received no transfusions during surgery. Actually, hemoglobin values recorded during surgery were probably overestimated due to the restrictive fluid therapy, thus masking the underlying anemia. We may therefore speculate that transfusing patients an earlier phase on the basis of DO_2_, rather than Hb levels, could improve oxygen delivery and prevent hyperlactatemia development [32,33].

This study has various limitations. First, this is a retrospective analysis performed in a single center. Second, the size of our study population is smaller than in other studies [2,3,4,6,8,13]. Therefore, the lack of evidence that some conditions such as diabetes mellitus or cirrhosis are not associated with hyperlactatemia in this study might be ascribed to the low number of cases included in the analysis. In addition, due to the small series of patients and to the low death rate, we could not provide information about the association between high lactatemia and mortality. Despite previous limitations, an advantage of this study was the inclusion of a cohort of patients who were homogeneously treated by the same team of surgeons and anesthesiologists.

## 5. Conclusions

This study shows an association between hyperlactatemia and reduced oxygen delivery, mainly caused by a relevant hemoglobin drop during surgery. Considering that postoperative high lactate levels predict severe complications, further studies are worth investigating if blood losses should be earlier corrected. Overall, non-invasive hemodynamic monitoring and in particular the DO_2_ recording in critical steps of liver surgery might be of great utility to prevent hyperlactatemia and possibly, subsequent complications.

## Figures and Tables

**Figure 1 biomedicines-10-01108-f001:**
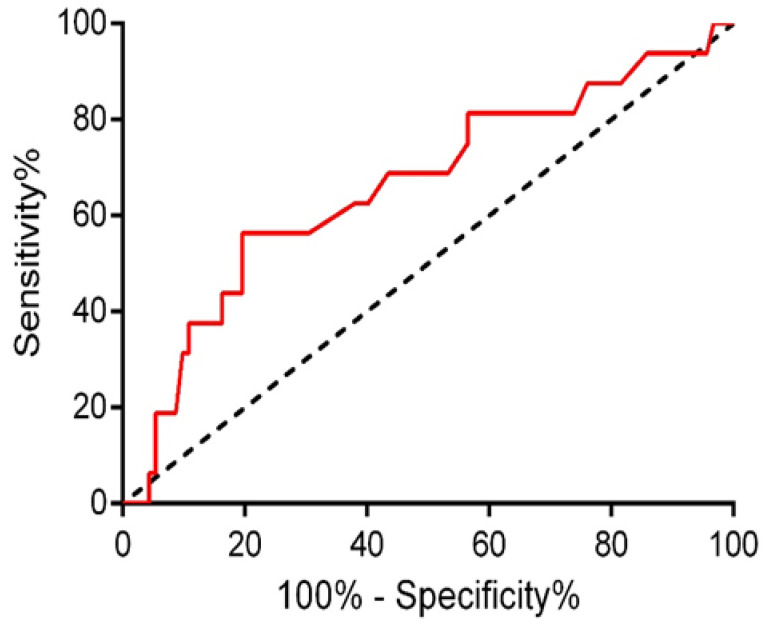
ROC curve analysis of post-operative lactate levels for the occurrence of severe complications. The optimal cut-off value was 5.05 mmol/L, with an area under the curve of 0.661 ± 0.079 (95% CI 0.506–0.816), a sensitivity of 0.563, and a specificity of 0.804.

**Figure 2 biomedicines-10-01108-f002:**
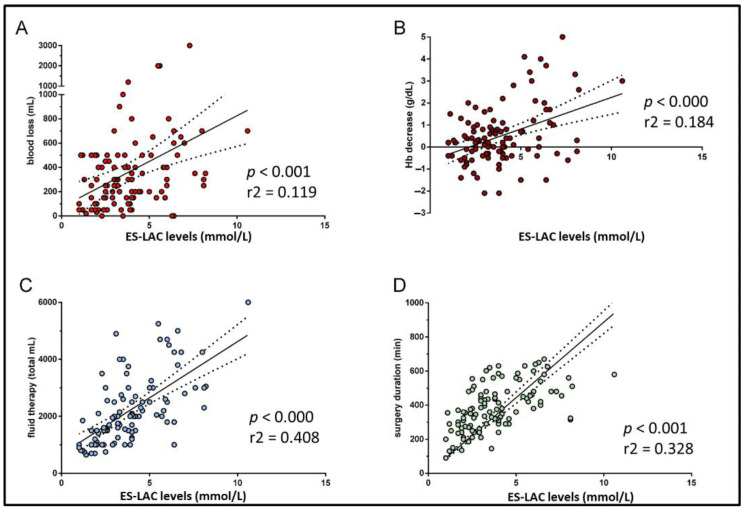
Linear regression analysis pairing lactate levels with drop of blood losses (**A**), Hb concentration (**B**), fluid administration (**C**), and surgery duration (**D**).

**Figure 3 biomedicines-10-01108-f003:**
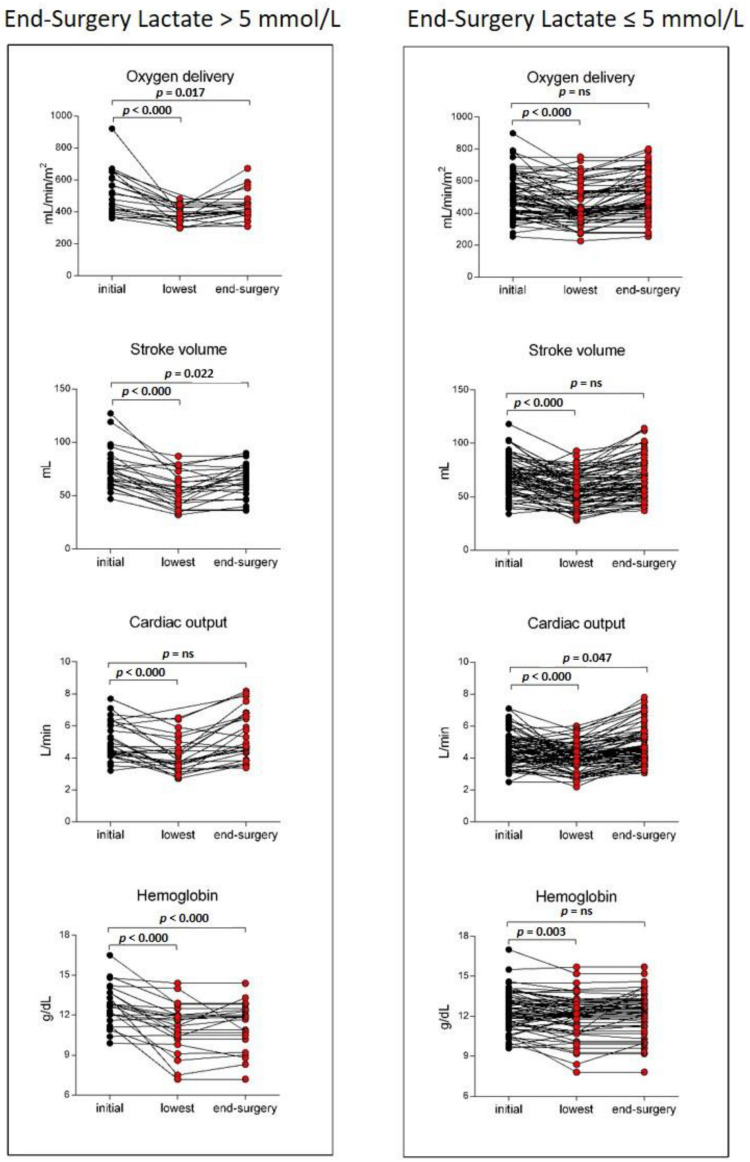
Paired analysis of initial, lowest, and end-surgery levels of oxygen delivery, stroke volume, cardiac output, and hemoglobin concentration in patients grouped according to end-surgery lactate level (≤ or >5 mmol/L). (ns, not significant).

**Table 1 biomedicines-10-01108-t001:** Clinical and surgery characteristics of the 108 investigated patients.

Patients	*n* = 108
** *Age, years, median (IQR)* **	62.0 (55.0–69.0)
** *Sex, male, No. (%)* **	62 (57.4)
** *BMI, kg/m^2^, median (IQR)* **	25.1 (22.1–28.1)
** *ASA classification, No. (%)* **	
1	3 (2.7)
2	88 (81.6)
3	17 (15.7)
** *Comorbidities, No. (%)* **	
Diabetes mellitus	9 (8.3)
Hypertension	54 (50)
Coronary artery disease	8 (7.4)
COPD	9 (8.3)
** *Indication for liver resection, No. (%)* **	
Liver metastasis	69 (63.9)
Hepatocellular carcinoma	21 (19.4)
Cholangiocellular carcinoma	5 (4.6)
Gallbladder cancer	4 (3.7)
Benign disease	3 (2.8)
Intra-hepatic stones	6 (5.6)
** *Underlying liver disease, No. (%)* **	
Cirrhosis	9 (8.3)
Fibrosis moderate/severe	19 (17.6)
Macrovescicular steatosis >25%	10 (9.3)
** *Surgery, No. (%)* **	
Laparotomy/Laparoscopy	94 (87.0)/14 (13.0)
Major liver resection	3.2 (29.6)
Minor resection	57 (52.8)
Minor complex resection	19 (17.6)
** *Overall ischemia time, min, median (IQR)* **	57 (28–85)
** *Duration of surgery, min, median (IQR)* **	375 (290–480)
** *Postoperative outcome* **	
Length of postoperative stay, days, median (IQR)	7 (6–11)
Clavien–Dindo complications (grade ≥ 3), No. (%)	16 (14.8)
90-Day postoperative mortality, No. (%)	1 (0.9)

BMI, body mass index; ASA, American Society of Anesthesiology Physiological Status; COPD, chronic obstructive pulmonary disease.

**Table 2 biomedicines-10-01108-t002:** Fluid balance and biochemical and hemodynamic parameters in the 108 investigated patients.

** *Fluids and transfusions* **	
Fluid administration, mL/kg/h, median (IQR)	3.9 (3.2–4.8)
Total intraoperative fluids, mL, median (IQR)	2000 (1325–2775)
Intraoperative blood loss, mL, median (IQR)	300 (150–500)
RBC transfused patients, No. (%)	10 (9.3)
Intraoperative diuresis, mL/h, median (IQR)	103 (79–131)
** *Hemodynamic parameters, median (IQR)* **	
Initial arterial lactate, mmol/L	1.1 (1.0–1.6)
End-surgery arterial lactate, mmol/L	3.6 (2.4–5.1)
Initial BE, mmol/L	4.0 (2.9–5.6)
End-surgery BE, mmol/L	0.8 (−1.1–2.5)
Initial pH	7.49 (7.45–7.52)
End-surgery pH	7.40 (7.36–7.45)
Initial blood glucose, mg/dL	93 (86–106)
End-surgery blood glucose, mg/dL	177 (151–217)
Initial Hb, g/dL	12.6 (11.5–13.4)
Lowest Hb, g/dL	12.0 (10.9–12.9)
End-surgery Hb, g/dL	12.3 (10.9–13.1)
Initial heart rate, btm	65 (59–77)
End-surgery heart rate, btm	76 (67–85)
Initial MAP, mmHg	86 (76–97)
Lowest MAP, mmHg	70 (62–78)
End-surgery MAP, mmHg	77 (69–83)
Initial SVV, %	10.0 (7.8–12.0)
Highest SVV, %	17.0 (13.0–19.0)
End-surgery SVV, %	12.0 (10.0–15.0)
Initial CO, L/min	4.5 (3.9–5.2)
Lowest CO, L/min	3.9 (3.5–4.5)
End-surgery CO, L/min	4.6 (3.9–5.7)
Initial DO_2_, mL/min/m^2^	487 (409–614)
Lowest DO_2_, mL/min/m^2^	413 (361–497)
End-surgery DO_2_, mL/min/m^2^	458 (400–585)
** *Noradrenaline administration, No. (%)* **	63 (58.3)

IQR, interquartile range; RBC, red blood cells; BE, actual base excess; Hb, hemoglobin; MAP, mean arterial pressure; SVV, stroke volume variation; CO, cardiac output; DO_2_, oxygen delivery.

**Table 3 biomedicines-10-01108-t003:** Demographics and clinical characteristics according to end-surgery arterial lactate concentration.

	*>5* mmol/L *n = 27*	*≤5* mmol/L *n = 81*	*p*-Value
** *Age, years, median (IQR)* **	61.0 (57.0–65.0)	63.0 (54.0–70.5)	0.563
** *Sex, male, No. (%)* **	17 (63.0)	45 (55.6)	0.654
** *BMI, kg/m^2^, median (IQR)* **	26.9 (24.1–30.8)	24.6 (21.8–27.8)	0.032
** *ASA classification, No. (%)* **			
1 2 3	0 25 (92.6) 2 (7.4)	3 (3.7) 63 (77.8) 15 (18.5)	0.419
** *Comorbidities, No. (%)* **			
Diabetes mellitus Hypertension Coronary artery disease COPD	2 (7.4) 19 (70.4) 0 2 (7.4)	7 (8.6) 35 (43.2) 8 (9.9) 7 (8.6)	1.000 0.025 0.197 1.000
** *Indication for liver resection, No. (%)* **			
Liver metastasis Hepatocellular carcinoma Cholangiocellular carcinoma Gallbladder cancer Benign disease Intra-hepatic stones	51 (63.0) 16 (19.8) 4 (4.9) 2 (2.5) 3 (3.7) 5 (6.2)	18 (66.7) 15 (18.5) 1 (3.7) 2 (7.4) 0 1 (3.7)	0.728
** *Underlying liver disease, No. (%)* **			
Cirrhosis Macrovescicular steatosis >25%	1 (3.7) 3 (11.1)	8 (9.9) 7 (8.6)	0.445 0.708
** *Surgery, No. (%)* **			
Laparotomy/Laparoscopy	26 (96.3)/1(3.7)	68 (84.0)/13(16.0)	0.182
Major resection Minor resection Minor complex resection	9 (33.3) 6 (22.2) 12 (44.4)	23 (28.4) 51 (63.0) 7 (8.6)	0.020
** *RBC postoperative transfused patients, No. (%)* **	7 (25.9)	3 (3.7)	0.002
** *Overall ischemia time, min, median (IQR)* **	83 (66–139)	57 (35–82)	0.001
** *Duration of surgery, min, median (IQR)* **	480 (440–587)	330 (267–432)	<0.001
** *Length of stay, days, median (IQR)* **	12 (7–17)	7 (5–9)	0.001
** *RBC transfused patients, No. (%)* **	8 (29.6)	9 (11.1)	0.032
** *Clavien–Dindo grade ≥3, No. (%)* **	9 (33.3)	7 (8.6)	0.004
** *90-day postoperative mortality, No. (%)* **	0	1 (0.9)	1.000

IQR, interquartile range; BMI, body mass index; ASA, American Society of Anesthesiology Physiological Status; COPD, chronic obstructive pulmonary disease; RBC, red blood cells.

**Table 4 biomedicines-10-01108-t004:** Biochemical, fluid balance, and hemodynamic parameters according to end-surgery arterial lactate concentration.

	*>5* mmol/L *n = 27*	*≤5* mmol/L *n = 81*	*p*-Value
** *Hemodynamic parameters, median (IQR)* **			
Initial arterial lactate, mmol/L	1.2 (1.1–2.0)	1.1(1.0–1.4)	0.039
End-surgery lactate level, mmol/L	6.4 (5.8–7.3)	3.0 (2.1–3.9)	<0.001
Initial BE, mmol/L	4.4 (3.6–5.5)	3.8 (2.9–5.7)	0.270
End-surgery BE, mmol/L	−1.0 (−4.2–0.6)	1.5 (−0.7–2.8)	<0.001
Initial pH	7.48 (7.42–7.51)	7.49 (7.45–7.52)	0.614
End-surgery pH	7.37 (7.35–7.40)	7.41 (7.38–7.45)	<0.001
Initial blood glucose, mg/dL	95.0 (86.0–111.0)	93.0 (85.5–104.0)	0.408
End-surgery blood glucose, mg/dL	200.0 (152.0–256.0)	170.0 (151.0–209.0)	0.130
Total intraoperative fluids, mL	3000 (2250–4375)	1575 (1100–2275)	0.001
Fluid administration, mL/kg/h	4.3 (3.8–5.3)	3.6 (3.0–4.8)	0.024
Intraoperative blood loss, mL	500 (250–500)	200 (100–400)	<0.001
Initial Hb, g/dL	12.8 (11.9–13.7)	12.4 (11.4–13.4)	0.458
Lowest Hb, g/dL	11.2 (10.2–12.0)	12.1 (11.2–13.0)	0.002
End-surgery Hb, g/dL	11.8 (10.0–12.4)	12.6 (11.4–13.2)	0.003
Initial heart rate, btm	67 (61–77)	65 (59–76)	0.575
End-surgery heart rate, btm	82 (76–90)	74 (64–83)	0.010
Initial MAP, mmHg	87 (76–105)	86 (76–95)	0.202
Lowest MAP, mmHg	66 (60–76)	71 (64–80)	0.142
End-surgery MAP, mmHg	75 (69–81)	77 (68–86)	0.421
Initial SVV, %	10 (7–12)	11 (8–12)	0.355
Highest SVV, %	18 (14–20)	16 (12–19)	0.184
End-surgery SVV, %	14 (11–16)	12 (10–15)	0.214
Initial SV, mL	72 (61–83)	70 (56–77)	0.286
Lowest SV, mL	53 (42–64)	55 (44–68)	0.592
End-surgery SV, mL	65 (54–76)	64 (52–82)	0.633
Initial CO, L/min	4.7 (4.2–6.1)	4.4 (3.8–5.1)	0.064
Lowest CO, L/min	3.8 (3.3–5.0)	3.9 (3.5–4.5)	0.951
End-surgery CO, L/min	4.7 (4.3–6.6)	4.4 (3.8–5.3)	0.064
Initial DO_2_, mL/min/m^2^	494 (409–624)	487 (407–599)	0.762
Lowest DO_2_, mL/min/m^2^	383 (338–432)	417 (375–532)	0.014
End-surgery DO_2_, mL/min/m^2^	410 (387–464)	488 (418–608)	0.008
** *Noradrenaline administration No. (%)* **	17 (63%)	28 (34.6%)	0.013

BE, actual base excess; RBC, red blood cells; Hb, hemoglobin; HR, heart rate; MAP, mean arterial pressure; SVV, stroke volume variation; SV, stroke volume; CO, cardiac output; DO_2_, Oxygen delivery.

**Table 5 biomedicines-10-01108-t005:** Correlation between ES-ALC and postoperative complications.

	*>5* mmol/L *n = 27*	*≤5* mmol/L *n = 81*	*p*-Value
Clavien–Dindo complications grade, No. (%)			
**Grade I** ** *Type of complications, No.* **	1 (3.7) Increase in serum creatinine (1)	1 (1.2) Pleural effusion (1)	0.439
**Grade II** ** *Type of complications, No.* **	9 (33.3) bile leak (1), anemia (7), coagulopathy (1), fever (3), abdominal collection (2), lymphatic fistula (1), pleural effusion (2), mild liver failure (2)	17 (21.0) bile leak (7), anemia (2) fever (6), abdominal collection (3), lymphatic fistula (1), pleural effusion (2), bowel obstruction (1), mild liver failure (1)	0.203
**Grade IIIa** ** *Type of complications, No.* **	7 (25.9) pleural effusion (1) abdominal collection (4), bile leak (2), sepsis (1), liver failure (1), nefrostomy (1)	3 (3.7) pleural effusion (1), abdominal collection (2), bile leak (1), ascites (1)	0.002
**Grade IIIb** ** *Type of complications, No.* **	1 (3.7) abdominal collection (1), pleural effusion (1)	1 (1.2) abdominal collection (1)	0.439
**Grade IVa** ** *Type of complications, No.* **	1 (3.7) dialysis (1)	2 (2.5) respiratory failure (1) gastric ulcer hemorrhage (1)	1.000
**Grade IVb**	0	0	
**Grade V** ** *Type of complications, No.* **	0	1 (1.2) Liver failure (1)	
**Pts with complications grade III-V**	9/27 (33.3)	7/81 (8.6)	0.004

**Table 6 biomedicines-10-01108-t006:** Univariate and Multivariate analysis for pre- and intraoperative factors associated with ES-ALC (>5 mmol/L) in 108 patients.

	Univariate Analysis		Multivariate Analysis	
	OR [95% CI]	*p*-Value	OR [95% CI]	*p*-Value
Age	0.99 (0.95–1.03)	0.621		
BMI	1.03 (1.00–1.06)	0.510		
Hypertension	3.12 (1.23–7.96)	0.017		
Overall ischemia time	1.02 (1.01–1.03)	0.001		
Duration of surgery	1.01 (1.01–1.02)	<0.001	1.011 [1.005–1.018]	0.001
Noradrenaline administration	3.22 (1.30–7.96)	0.011		
RBC transfused patients	9.10 (2.16–36.37)	0.003		
Initial arterial lactate	2.42 (1.11–5.26)	0.026		
End-surgery BE	0.61 (0.48–0.76)	<0.001		
Blood loss	1.00 (1.00–1.00)	0.004		
Total fluid administration	1.00 (1.00–1.00)	<0.001		
Lowest Hb	0.83 (0.66–1.05)	0.126		
End-surgery Hb	0.67 (0.51–0.88)	0.004		
Lowest DO_2_	0.99 (0.99–0.99)	0.011	0.989 [0.979–0.999]	0.028
End-surgery DO_2_	0.99 (0.99–1.00)	0.206		

OR, odds ratio; 95% CI, 95% confidence intervals; BMI, body mass index; RBC, red blood cells; BE, actual base excess; Hb, hemoglobin; DO_2_, Oxygen delivery.

## Data Availability

Not applicable.

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
