# Peer review of "High Arterial Lactate Levels after Hepatic Resection Are Associated with Low Oxygen Delivery and Predict Severe Postoperative Complications"

_biomedicines, 2022, doi:10.3390/biomedicines10051108_

Round 1

Reviewer 1 Report

The topic is not new but considering the several factors analyzed, some important corrections would be necessary:

  1. The base of statistic is only 27 patients, patient with ES-ALC >5.0 mmol/L and this is very small sample to build a regression analysis. The sample needs to be implemented.
  2. It needs a table with the list of the main complications: renal injury, pulmonary complications, bile leak,… This to understand why high ES-ALC might be correlated with the complications. More comment is needed on this part.
  3. Same for mortality. Why did they die? Liver failure, cardiac event, sepsis,..
  4. It’s important to mention how the anesthetist corrected a low pH and the authors should correlate the use of bicarbonate with the high ES-ACL

Author Response

REPLY TO REVIEWER #1:

We thank the Reviewer for his detailed analysis of our manuscript and for his constructive criticisms and advices. We have modified the manuscript according to his comments and questions.

All the new inclusions and changes performed in the text following the comments are highlighted in yellow.

Comment: The topic is not new but considering the several factors analyzed, some important corrections would be necessary:

1. The base of statistic is only 27 patients, patient with ES-ALC >5.0 mmol/L and this is very small sample to build a regression analysis. The sample needs to be implemented.

Response: we agree with the reviewer that the sample may be small for the analysis. However, as showed in the new table 6, our study population allowed us to develop a significant and strong multivariable model. We better described the results of the multivariable logistic regression analysis (page 11, lines 304-312) and we included the new table 6. A comment has been included on page 14, lines 379-381.

2. It needs a table with the list of the main complications: renal injury, pulmonary complications, bile leak,… This to understand why high ES-ALC might be correlated with the complications. More comment is needed on this part.

Response: a new table (table 5) has been included in the results section lines 278-283.

3. Same for mortality. Why did they die? Liver failure, cardiac event, sepsis,..

Response: cause of postoperative mortality has been included in the new table 5.

4. It’s important to mention how the anesthetist corrected a low pH and the authors should correlate the use of bicarbonate with the high ES-ACL.

Response: a new sentence about the anesthesiologist management of low pH has been included on page 3, lines 119-122 and lines 126-130. A comment about the low pH has been included in the results section on page 7 lines 258-260.

Reviewer 2 Report

Pretty cool paper. Well written. I would recommend it's publication.

Author Response

REPLY TO REVIEWER #2: 

Comment: Pretty cool paper. Well written. I would recommend it's publication.

Response: We thank the Reviewer for his comment.

Reviewer 3 Report

In the manuscript titled “High Arterial Lactate Levels after Hepatic Resection are Associated with Low Oxygen Delivery and Predict Severe Postoperative Complications” Gaspari R et al. present a retrospective analysis of 108 patients who underwent elective liver resection. High End-Surgery Arterial Lactate Concentration might predict poor outcome after surgery, therefore authors tried to identify intraoperative hemodynamic parameters predicting high End-Surgery Arterial Lactate Concentration during elective liver resection. The topic is interesting and of high importance, and the manuscript is well written. However, I have some comments/questions:

  1. First, for me it is not totally clear what was the focus of the study - high arterial lactate level and its association with outcomes or measurements with FloTrac/EV1000 TM.
  2. Patients for the analysis were operated in 2017-2018. As it is now 2022, was it not possible to include more patients also from 2019-2021?
  3. Two groups of patients (high vs low lactate) are compared. However, in a half of the patient’s minor liver resections were performed. It might be speculated that in these patients’ evaluation of lactate levels is less important. Did authors compare minor vs major liver resections? Laparoscopic vs open liver resections?
  4. Tables should be better formatted. Headings/subheading might make tables easier to read.

Author Response

REPLY TO REVIEWER #3: 

We thank the Reviewer for the analysis of our manuscript and for his comments and advices. We have modified the manuscript according to his questions.

All the new inclusions and changes performed in the text following the comments are highlighted in yellow.

Comment: In the manuscript titled “High Arterial Lactate Levels after Hepatic Resection are Associated with Low Oxygen Delivery and Predict Severe Postoperative Complications” Gaspari R et al. present a retrospective analysis of 108 patients who underwent elective liver resection. High End-Surgery Arterial Lactate Concentration might predict poor outcome after surgery, therefore authors tried to identify intraoperative hemodynamic parameters predicting high End-Surgery Arterial Lactate Concentration during elective liver resection. The topic is interesting and of high importance, and the manuscript is well written. However, I have some comments/questions:

1. First, for me it is not totally clear what was the focus of the study - high arterial lactate level and its association with outcomes or measurements with FloTrac/EV1000 TM.

Response: we better defined the end-points of our study on page 4, lines 181-185.

2. Patients for the analysis were operated in 2017-2018. As it is now 2022, was it not possible to include more patients also from 2019-2021? 

Response: the time period of this specific analysis is correlated with the same anesthesiologist and surgical team. Members of this team changed after 2018. For this reason we did not included patients resected after 2018, by a different anestehesiologist management, because they could represent a bias for the evaluation of the results.

3. Two groups of patients (high vs low lactate) are compared. However, in a half of the patient’s minor liver resections were performed. It might be speculated that in these patients’ evaluation of lactate levels is less important. Did authors compare minor vs major liver resections? Laparoscopic vs open liver resections? 

Response: thank you for this important question. In table 3 we showed that the rate of open resections and of laparoscopic resections were not significantly different in the two groups. According to the extent of liver resection, in this analysis, we did not evaluate only major vs. minor resections. Indeed, minor complex liver resections were also included in table 3. Minor complex liver resections are minor resections but they include multiple parenchymal sparing liver resections that are considered as the gold standard for the treatment of colorectal liver metastases. They are complex interventions often associated with long duration of surgery and with long duration of pedicle clamping. They represent an interesting model to evaluate the incidence of high ES-ALC. In this series minor complex resections were performed in about 18% of patients and they were significantly associated with high ES-ALC.

4. Tables should be better formatted. Headings/subheading might make tables easier to read.

Response: we better formatted the tables with headings and subheading in bold. We also included the abbreviations at the end of each table.